# Lean Six Sigma Approach to Improve the Management of Patients Undergoing Laparoscopic Cholecystectomy

**DOI:** 10.3390/healthcare12030292

**Published:** 2024-01-23

**Authors:** Arianna Scala, Giovanni Improta

**Affiliations:** 1Department of Public Health, University of Naples “Federico II”, 80138 Naples, Italy; ing.improta@gmail.com; 2Interdepartmental Center for Research in Healthcare Management and Innovation in Healthcare (CIRMIS), University of Naples “Federico II”, 80138 Naples, Italy

**Keywords:** laparoscopic cholecystectomy, DMAIC cycle, health management, post-operative length of stay

## Abstract

Laparoscopic cholecystectomy (LC) is the gold standard technique for gallbladder diseases in both emergency and elective surgery. The incidence of the disease related to an increasingly elderly population coupled with the efficacy and safety of LC treatment resulted in an increase in the frequency of interventions without an increase in surgical mortality. For these reasons, managers implement strategies by which to standardize the process of patients undergoing LC. Specifically, the goal is to ensure, in accordance with the guidelines of the Italian Ministry of Health, a reduction in post-operative length of stay (LOS). In this study, a Lean Six Sigma (LSS) methodological approach was implemented to identify and subsequently investigate, through statistical analysis, the effect that corrective actions have had on the post-operative hospitalization for LC interventions performed in a University Hospital. The analysis of the process, which involved a sample of 478 patients, with an approach guided by the Define, Measure, Analyze, Improve, and Control (DMAIC) cycle, made it possible to reduce the post-operative LOS from an average of 6.67 to 4.44 days. The most significant reduction was obtained for the 60–69 age group, for whom the probability of using LC is higher than for younger people. The LSS offers a methodological rigor that has allowed us, as already known, to make significant improvements to the process, standardizing the result by limiting the variability and obtaining a total reduction of post-operative LOS of 67%.

## 1. Introduction

Laparoscopic cholecystectomy (LC) is the gold standard technique for gallbladder diseases in both emergency and elective surgery [1]. LC results in a lower overall complication rate and a shorter post-operative hospital stay compared to open cholecystectomy. In situations where LC is dangerous, a surgeon may be forced to change from laparoscopy to an open procedure. Literature data show that 2 to 15% of laparoscopic cholecystectomies are converted to open surgery during surgery for various reasons [2,3]. The most common causes are peritoneal adhesions and inflammatory infiltration of the gallbladder [4,5]. Converted cases are associated with an increased number of infections and other post-operative complications [6], an increased risk of additional procedures, and a higher rate of readmission within 30 days [7]. Additionally, the conversion from laparoscopic to open surgery results in longer post-operative stays and higher morbidity and mortality rates in this group of patients [8].

The modern literature points to another alarming phenomenon. Pogorelić et al. [9] have shown a significant growth in pediatric cholecystectomies over the last twenty years. As demonstrated by the authors and confirmed by several other studies, this growth is attributable to the increase in the prevalence of childhood obesity [10,11,12], showing a correlation between weight increment and the presence of symptomatic cholelithiasis.

LC is the gold standard technique for gallbladder diseases in both emergency and elective surgery [13] and also in pediatric surgery, showing short post-operative LOS with low complication rates [1]. LC is a minimally invasive procedure with no specific contraindications except for a few surgical risk factors [14]. The incidence of the disease related to an increasingly elderly population, coupled with the efficacy and safety of LC treatment [15], has resulted in an increase in the frequency of interventions without an increase in surgical mortality [16], making LC the most common procedure in general surgery [17]. Although it has many advantages, such as a shorter hospital stay and convalescence [18,19], conversion, however, is associated with increased overall morbidity, surgical site and lung infections, and longer hospital stays [20,21]. From an economic point of view, therefore, LC is associated with savings linked to the post-operative period [22], which is opposed to a higher expense in terms of the complexity of the equipment and disposable material [23]. As with any surgery, much of the total costs can be attributed to consumables, and it is essential that healthcare organizations understand the costs of high-volume surgical procedures, especially in an era where healthcare spending is a concern for many nations in the world [24,25,26]. Hospital managers continually seek to reduce the cost of patient care on the one hand and to maintain or improve the quality of patient care on the other. Governments also define national strategies using efficiency and performance indicators. Specifically, the Italian Ministry of Health has defined, in the National Outcome Plan (in Italian PNE) [27], an outcome indicator that analyzes the post-operative length of stay (LOS) of LC interventions to evaluate the performance of the structure. In fact, a post-operative LOS within 3 days is set as the standard, which must be guaranteed for at least 70% of LC interventions [28]. To accomplish this, it is essential to increase the information associated with that particular health process. With more information, it is possible to organize optimally to deliver the best results. With this in mind, data collection becomes strategic for each organization [29]. A multitude of methodologies with respect to conducting data analysis have been recorded in scholarly works. These techniques empower researchers and healthcare experts to derive invaluable insights from biomedical information, facilitating the foundation of decisions on evidence and enhancing the efficiency of healthcare provision. One of the techniques that has revolutionized healthcare management is Lean Six Sigma (LSS). LSS is a process improvement method that aims to increase uniformity and operational quality and reduce variation and waste [30]. It has been developed and is widely used in industry and has recently been introduced, on a limited scale, in the healthcare sector. The healthcare organization is the place where defects and errors cannot be tolerated. A simple mistake can cost human life, so flaws or mistakes must be eliminated [31]. For this reason, an approach with the characteristics of the LSS becomes strategic. Specifically, the Lean approach evaluates a process from the point of view of finding and removing work that does not add value, promoting the better execution of those steps of the process that do add value; while Six Sigma approaches are based on precise measurements, accurate process variables, and analyses of the results designed to find those factors that most affect customer/patient satisfaction [32]. For example, Bhat et al. (2016) showed the benefits of an LSS approach for the reduction of turn-around time in the medical record preparation process [33]. From a purely welfare point of view, Mandahawi et al. (2011) showed how an LSS approach, and in particular the use of the DMAIC cycle, reduced the hospital stay (LOS) of patients requiring ophthalmic service by 48%, while Improta et al. (2019) showed how the corrective actions identified by the LSS analysis of the fast track process of hip surgery implemented by the University Hospital “Federico II” of Naples (Italy) led to a reduction of more than 25% of the LOS [34]. The aim of this study was to exploit the advantages of an LSS approach to identify and subsequently investigate the effect that corrective actions have had on post-operative hospitalization for LC interventions performed at the University Hospital “San Giovanni di Dio e Ruggi d’Aragona” of Salerno (Italy). This work aims to extend our previous work, in which the criticalities of the process and the benefits obtained were shown both for a limited number of years and a reduced set of variables [35]. The same technique has already brought significant benefits (over 39%) in reducing pre-operative hospitalization for patients who enter the hospital with a femur fracture [36].

## 2. Materials and Methods

In this retrospective work, the data of patients who underwent LC at the University Hospital “San Giovanni di Dio e Ruggi d’Aragona” of Salerno (Italy) were analyzed. The aim is to reduce post-operative LOS according to the indications of the Italian government. An LSS approach and, in particular, the implementation of the DMAIC cycle have been used. In this way, all the issues and improvement strategies have been identified.

A 10-year sample was used to better identify critical issues and study the benefits of the improvement strategies. In particular, data were collected retrospectively and the dataset of 478 patients was divided into two groups based on the year of discharge: 5 years before (2010–2015), consisting of the information of 254 patients; and 5 years after the first implementation of the DMAIC cycle (2016–2020), consisting of the demographic and clinical data of 224 patients. For each patient, the following information was extracted from the QuaniSDO hospital information system, a type of software used at the hospital for the management of discharge forms:Gender (Male/Female);Age (0–29, 30–49, 50–59, 60–69, 70–79, ≥80);Hypertension (Yes/No);Diabetes (Yes/No);Cardiovascular disease (Yes/No);Respiratory disease (Yes/No);Obesity (Yes/No);Liver disease (Yes/No);Date and time of admission;Date and time of an LC procedure;Date and time of discharge.

This study has as its primary outcome the evaluation of the corrective actions implemented on the pathway of the patient who is admitted for LC in view of LSS. In addition to this, the classification of patients according to age, gender, and a selected group of comorbidities (as shown in the list above) makes it possible, as secondary outcomes, to identify the distribution of patients in these sub-groups and the criticalities associated with particular conditions in order to study ad hoc corrective solutions and, finally, to assess the impact of these actions on a class-by-class level.

### 2.1. DMAIC Cycle

The DMAIC cycle is a well-structured methodology used not only to identify the issues or the causes of inefficiencies but also to improve the process quality. DMAIC is an acronym for the 5 phases that constitute it: Define, Measure, Analyze, Improve, and Control. Figure 1 shows a generic description of the cycle’s phases.

Unlike the Deming cycle (Plan–Do–Check–Act), the DMAIC includes the application of statistical tools to perform data analysis. The five phases are presented in detail in the following paragraphs.

#### 2.1.1. Define

In the first phase of the LSS improvement process, as reported in Figure 1, the project team drafts a project charter, i.e., a high-level project definition. In this case, the objective is “reduction of post-operative LOS for patients undergoing LC”, and the critical to quality (CTQ) is the post-operative LOS. The following information is reported in the project charter:Project title: Lean Six Sigma approach to improve the management of patients undergoing laparoscopic cholecystectomy.Question: prolonged post-operative LOS.CTQ: post-operative LOS (days).Target: Improve the quality process to reduce the CTQ for all patients who undergo LC according to the indication of the Italian government.Deliverables: increase in the turnover; decrease in post-operative LOS.Timeline:○Define: January–March 2015;○Measure: April–August 2015;○Analyze: September–October 2015;○Improve: November–December 2015;○Control: January 2016–December 2020.In scope: laparoscopic cholecystectomy; transversal to surgery department of “San Giovanni di Dio e Ruggi d’Aragona” of Salerno (Italy).Out scope: the departments not enclosed in the process.Financial: no financial assistance is required.Business need: speeding up the discharge process.

#### 2.1.2. Measure

In this phase, the as-is clinical process was measured. A retrospective measurement of the CTQ was carried out to obtain a significant trend in the LOS. For this reason, starting from the diagnosis-related group (DRG) according to the inclusion/exclusion criteria in the measurement protocol for the characterization of the PNE indicator, the information of 254 patients discharged in the five year period (2010–2015) was extracted from QuaniSDO. In particular, all patients who underwent LC with a diagnosis of gallbladder or bile duct lithiasis were included. On the other hand, the following were excluded: pediatric patients, patients who were not resident in Italy, pregnant women, patients who had been transferred, and deceased patients or those who were hospitalized for other reasons (such as trauma, malignant abdominal tumor, for other abdominal surgery, or for an open procedure).

The difference in days between the date and time of LC execution and the date and time of discharge is the post-operative LOS. Descriptive statistics are reported in Table 1, while Figure 1 shows the run chart. The statistical analysis was obtained using IBM SPSS Statistics Version 26.0 software (IBM Corp, Armonk, NY, USA), while the graphs were obtained with the support of Microsoft Excel (MS Office 2016 suite) (Microsoft Corp, Redmond, WA, USA).

#### 2.1.3. Analyze

At this juncture, every facet of the initial post-operative process was delineated and thoroughly deliberated upon by the multidisciplinary team, made up of different personalities, such as doctors, nurses, and engineers, and led by the healthcare director, who was responsible for the analysis. The proceedings were overseen by the hospital’s healthcare director and their team. Subsequently, an exploration into the fundamental triggers behind extended post-operative length of stay (LOS) was undertaken, facilitated by an Ishikawa diagram. The diagram concentrated on two primary dimensions: “People” and “Process”. These dimensions were further subdivided into four distinct categories: “Patients” and “Clinical staff” were categorized under the “People” dimension, while “Hospital” and “Process” were categorized under the “Process” dimension. Specifically, the category “Clinical staff” revealed staff resistance to change and a lack of interest in the economic aspects of the process; while as far as “Hospital” is concerned, there is an excessively high waiting time for post-operative examinations; then, for “Process” there is a lack of a standardized discharge procedure and thus of a diagnostic therapeutic care pathway. Finally, with regard to the category “Patients”, a statistical analysis was carried out to investigate how post-operative LOS changes according to clinical and demographic factors. The variables considered were age, sex, hypertension, diabetes, cardiovascular disease, respiratory disease, obesity, and liver disease. To identify the most appropriate statistical tests with which to characterize the population under consideration, the distribution was tested in the preliminary phase using the Kolmogorov–Smirnov test. The result shows the non-normality of the distribution, which implies the use of nonparametric tests. The analysis, therefore, was carried out using two tests: the U-Mann–Whitney test for dichotomous variables and the Kruskal–Wallis test for non-dichotomous ones. A 95% confidence level was set; therefore, the test was considered to be verified when the *p*-value was <0.05.

#### 2.1.4. Improve

From the discussion started in the previous phase, combined with the results obtained from data processing and from all the literature available on the process in question, the criticalities of the process, which begins when the patient is admitted to the hospital until his discharge, were identified. The multidisciplinary nature of the team and the capillarity of the description make it possible to identify the possible corrective strategies to be implemented to increase the quality of the process, with positive effects also in contrasting the prolongation of the CTQ. In particular, a defined pathway was established for the post-operative course and discharge. The steps to be followed are as follows:Post-operative hematochemical examinations: the time for performing and receiving the result has been reduced thanks to a defined pathway with the laboratories.Light feeding after gas canalization.Discharge on the first or second day depending on clinical evolution.Outpatient surgical check-up 7 days after discharge.

### 2.2. Control

In this phase, that of Control, the improvement actions identified in the Improve phase are standardized and confirmed. To do this, we decided to analyze the post-operative LOS of the 224 patients who underwent LC surgery in the 5 years following the conclusion of the first implementation of the DMAIC cycle (2016–2020). To compare this process with the as-is process, the same set of demographic and clinical variables that influence the CTQ already used in the Analyze phase was studied. In this way, it was possible to use statistical tests, and in particular the U-Mann–Whitney test at a significance level α = 0.05, to directly compare the beneficial effects of corrective actions on the total and on all categories of patients identified. Another run chart was created using Microsoft Excel (MS Office 2016 suite) (Microsoft Corp, Redmond, WA, USA) to graphically display the reduction obtained.

## 3. Results

First, we decided to display the status in graphical form using a run chart. Figure 2 reports the change in post-operative LOS in days for all 254 observations relating to the 5-year period 2010–2015. The red line shows the average value of 6.57 days.

From a careful analysis of the process, especially the phase leading to discharge, four main causes have been identified that have produced an extension of LOS. The Ishikawa diagram was used for the analysis, as shown here in Figure 3.

From this preliminary analysis and from all subsequent meetings, the project team highlighted that the variability in the discharge process generated a delay that caused an inevitable increase in post-operative LOS. Therefore, a defined pathway was established for the post-operative course and discharge. Statistical analysis was conducted to understand how patient demographic and clinical factors affect hospitalization. The results are reported in Table 1.

The results show that no variables significantly affect (*p*-value < 0.05) the LOS. Furthermore, for all the classes highlighted, the post-operative LOS settles at an average value higher than 6 days, in any case higher than the 3 days foreseen by the Italian government. Some conditions, such as patients suffering from liver disease or patients under the age of 30, are associated with a higher, although not significant, average LOS.

In the improvement phase, the information obtained was processed by the project team for the identification of corrective actions. Although no previous demographic and clinical conditions of the patient statistically influenced the LOS, the discharge process was revised for all patients to arrive at a standardized and shared form.

Starting from the adoption of the corrective measures identified, the data of 224 subjects who were hospitalized in the years 2015–2020 were analyzed using the run chart (Figure 4).

There was a significant reduction in post-operative LOS, which went from over 6 days to over 4 days. Although still far from the predicted value in the PNE, a reduction of over 67% was achieved. At this point, we decided to compare what happens for the categories of patients identified pre- and post-improvement using statistical analysis (Table 2).

The statistical analysis shows that for both women and men, there is a significant reduction in post-operative LOS of over 20% in the first case and over 40% in the second. Significant reductions are also obtained for patients with age group 60–69, hypertensive or not, diabetic or not, and with or without cardiovascular diseases. A reduction of approximately 30% was also obtained for non-obese and non-liver disease patients. Unlike the others, for patients in the age group 30–49, there was an increase in the LOS. From this result, it is clear that specific corrective actions and paths must be carried out for patients according to age.

## 4. Discussions

In this work, the data of 478 patients who underwent LC surgery at “San Giovanni di Dio e Ruggi d’Aragona” University Hospital of Salerno (Italy) in 2010–2020 were processed. Specifically, in 2016, the hospital adopted corrective measures, such as the standardization of the discharge process, to reduce the post-operative LOS for all patients who underwent LC, in accordance with the provisions of national guidelines. To analyze flow-related problems, an LSS approach and, in particular, the DMAIC cycle was implemented. Tools such as the run chart, Ishikawa diagram, and statistical analysis were used to support the Measure, Analyze, and Control phases to determine, with the use of objective data and brainstorming, the criticalities of the path. The dataset was divided into two groups with a separation threshold set on 31 December 2015. The variables considered, such as gender, age, hypertension (Yes/No), diabetes (Yes/No), cardiovascular disease (Yes/No), respiratory disease (Yes/No), obesity (Yes/No), and liver disease (Yes/No), allowed for the division of the dataset into homogeneous categories and a more precise analysis of the problem. All the data relating to the patients undergoing LC surgery in the years 2016–2020 therefore represent the dataset used to show and subsequently confirm the benefits obtained. There was a significant difference for both men and women, for patients with or without hypertension, for patients with and without cardiovascular disease, for patients without respiratory or liver disease, and for non-obese patients.

Several studies in the literature have tried to optimize the pathway of the LC patient. Ryan et al. [37] point out in their review that many authors have shown how pathway creation produces a reduction in LOS. For example, Soria et al. [38] reported a reduction of one day in hospital stay that resulted in a cost saving of approximately EUR 300, while Uchiyama et al. [39] created a unified pathway for patients undergoing laparoscopic surgery and still obtained a benefit from LOS values more similar to those shown in our study. With regard to the comorbidities under consideration, the literature review shows that certain parameters have a greater influence on the increase in LOS. For example, Ko-Iam et al. [40] do not show a significant difference in terms of age, sex, cardiovascular disease, and hypertension, whereas history of cirrhosis seems to have an influence. Liver problems, if not properly treated, can in fact generate complications in the post-operative period that prolong the patient’s hospital stay. For respiratory disorders, the literature also shows a correlation with LOS, as well as with mortality risks and the use of supportive devices [41]. The patient’s weight is a different matter, which in several works does not seem to affect LOS [42,43], even though our corrective actions only had beneficial effects for non-obese patients.

Although several works deal with LC, a direct comparison cannot be made as our study, not having used medical records, does not allow us to characterize comorbidities punctually or to include variables, such as operative time [44], which strongly impact on hospital stay. Furthermore, the observation does not include the last few years and also does not characterize the dataset according to month or day of the week. In addition to these limitations, this study is monocentric and carried out in a specific hospital, so the pathway and benefits obtained are not generalizable.

However, the study offers a tool that proves to be valid in this area and which can easily be implemented in other settings. The LSS and, in particular, its DMAIC cycle as developed in this article can be used as a guide for studying both similar and different pathways and objectives [45]. However, its practical implication in this study, which resulted in a reduction of LOS, remains valid. In addition, the criticalities highlighted in this study could also be encountered in other hospitals, thus already offering a starting point for the resolution of criticalities.

Further studies of the process will need to be conducted to understand what change produced this adverse effect. However, the benefit obtained is still more relevant given the epidemiology of the disease which, as is known, involves the older population more [46].

## 5. Conclusions

In this work, born as an extension of a previous publication, the LSS approach, and particularly the DMAIC cycle, was used to analyze and improve the process followed by patients who undergo LC at “San Giovanni di Dio e Ruggi d’Aragona” University Hospital of Salerno (Italy). With the implementation of this rigorous methodology, post-operative LOS dropped from an average of 6.67 days to 4.44 days. In contrast to the pertinent literature pertaining to comparable clinical routes, the suggested investigation exhibited intriguing and significant findings, offering a valuable roadmap for enhancing the quality of studies in healthcare environments. Nevertheless, as indicated by the literature, it is crucial to emphasize that the inherent features and circumstances of the institution where the clinical pathway is put into practice should be given careful attention, as they may exert a considerable impact on the results and consequences of the process.

## Figures and Tables

**Figure 1 healthcare-12-00292-f001:**
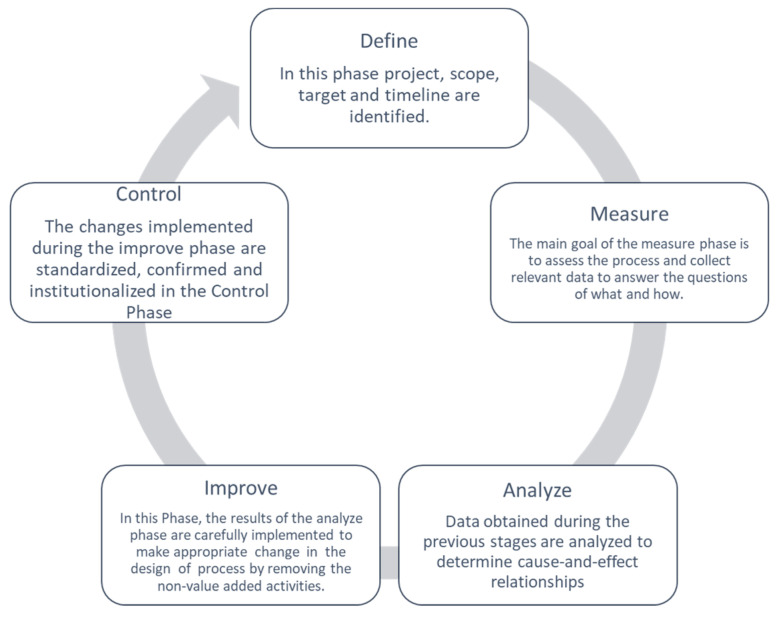
DMAIC cycle.

**Figure 2 healthcare-12-00292-f002:**
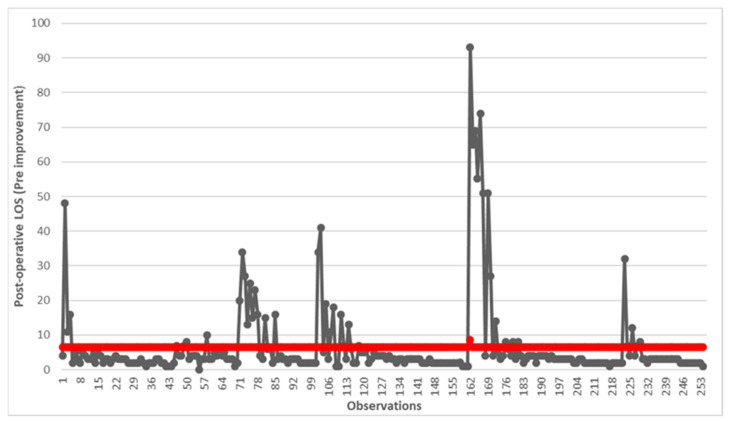
Run chart of post-operative LOS before the implementation of improvements. The red line indicates the average value of 6.57 days.

**Figure 3 healthcare-12-00292-f003:**
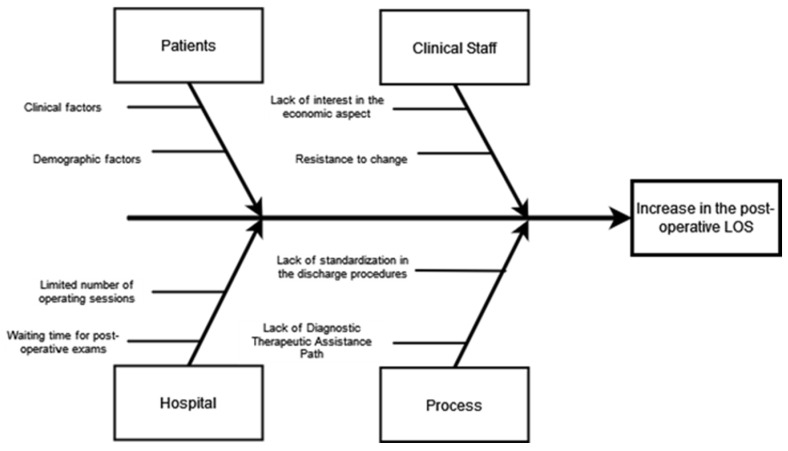
Ishikawa diagram.

**Figure 4 healthcare-12-00292-f004:**
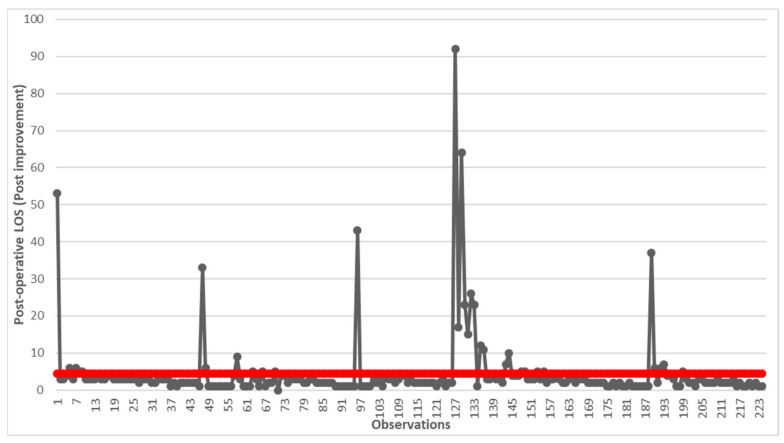
Run chart of post-operative LOS after the implementation of improvements. The red line indicates the average value of 4.44 days.

**Table 1 healthcare-12-00292-t001:** Descriptive statistics (AS-IS).

Variable	Category	Post-Operative LOS * (Mean ± Std Dev **)	N	*p*-Value
Age	0–29	10.000 ± 20.008	13	0.382
	30–49	6.655 ± 9.913	55	
	50–59	6.250 ± 12.128	68	
	60–69	6.592 ± 13.681	71	
	70–79	6.122 ± 8.838	41	
	over 80	4.833 ± 4.535	6	
Gender	Man	6.780 ± 12.910	104	0.886
	Women	6.430 ± 11.380	150	
Hypertension	No	6.727 ± 11.063	143	0.527
	Yes	6.369 ± 13.174	111	
Diabetes	No	6.638 ± 11.953	213	0.540
	Yes	6.220 ± 12.433	41	
Cardiovascular disease	No	6.694 ± 12.935	180	0.426
	Yes	6.270 ± 9.453	74	
Respiratory disease	No	6.242 ± 11.174	231	0.283
	Yes	9.870 ± 18.450	23	
Obesity	No	6.634 ± 12.368	227	0.386
	Yes	6.037 ± 8.555	27	
Liver disease	No	6.304 ± 11.414	227	0.516
	Yes	8.815 ± 16.272	27	

* LOS: length of stay; ** Std Dev: standard deviation.

**Table 2 healthcare-12-00292-t002:** Comparative and statistical analysis before and after the corrective actions.

Variable	Category	Pre-Improvement (Mean ± Std Dev *)	Post-Improvement (Mean ± Std Dev *)	Difference of the Mean (%)	*p*-Value
All	All	134	23.80	5	262
Age	0–29	10.000 ± 20.008	1.750 ± 0.500	−82.50	**0.008**
	30–49	6.655 ± 9.913	7.159 ± 17.430	+7.57	**0.002**
	50–59	6.250 ± 12.128	4.235 ± 6.752	−32.24	0.193
	60–69	6.592 ± 13.681	2.950 ± 3.730	−55.25	**0.0001**
	70–79	6.122 ± 8.838	4.038 ± 7.082	−34.04	0.086
	over 80	4.833 ± 4.535	5.307 ± 5.907	+9.80	0.858
Gender	Man	6.780 ± 12.910	3.906 ± 7.049	−42.39	**0.001**
	Women	6.430 ± 11.380	4.836 ± 10.825	−24.79	**0.0007**
Hypertension	No	6.727 ± 11.063	4.396 ± 9.134	−34.65	**0.0001**
	Yes	6.369 ± 13.174	4.504 ± 9.731	−29.28	**0.017**
Diabetes	No	6.638 ± 11.953	4.511 ± 9.846	−32.04	**0.0001**
	Yes	6.220 ± 12.433	9.900 ± 15.933	−16.48	**0.044**
Cardiovascular disease	No	6.694 ± 12.935	4.380 ± 8.381	−34.56	**0.0001**
	Yes	6.270 ± 9.453	4.000 ± 8.367	−36.20	**0.012**
Respiratory disease	No	6.242 ± 11.174	4.513 ± 9.984	−27.69	**0.0001**
Cardiological Disorders	Yes	9.870 ± 18.450	4.269 ± 5.204	−56.74	0.119
Obesity	No	6.634 ± 12.368	4.465 ± 9.490	−32.69	**0.0001**
	Yes	6.037 ± 8.555	4.522 ± 9.686	−28.30	0.125
Liver disease	No	6.304 ± 11.414	4.432 ± 9.644	−29.69	**0.0001**
	Yes	8.815 ± 16.272	5.700 ± 7.718	−35.33	0.245
Hypertension	No	6.727 ± 11.063	4.396 ± 9.134	−34.65	**0.0001**
	Yes	6.369 ± 13.174	4.504 ± 9.731	−29.28	**0.017**

* Std Dev: Standard Deviation. In bold are *p*-values < 0.05 (statistically significant).

## Data Availability

The data presented in this study are available from the authors upon reasonable request.

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
