# Peer review of "Lean Six Sigma Approach to Improve the Management of Patients Undergoing Laparoscopic Cholecystectomy"

_healthcare, 2024, doi:10.3390/healthcare12030292_

Round 1

Reviewer 1 Report

Comments and Suggestions for Authors

The authors investigated the methodological approach of Lean Six Sigma and aimed to determine the impact of corrective action on postoperative hospital stay in laparoscopic cholecystectomy procedures in a University hospital and then examined it through statistical analysis

I read the study with great interest. The study is well designed and of interest to readers. The discussion is the weakest point of the manuscript and should be significantly improved. My suggestions for improvement are as follows:

1. Abstract – The authors have used several abbreviations. Each abbreviation in the abstract should be listed in the full title the first time it is mentioned (e.g., LSS, DMAIC).

2. In the introduction, the authors correctly state that cholelithiasis is most common in women and in the elderly population. In recent decades, the incidence of gallstone disease in children has increased, primarily related to the epidemic of pediatric obesity. A recent study has shown that the number of pediatric cholecystectomies has increased significantly over the last 20 years, as has the average BMI of the population observed. This probably indicates a link between the rising obesity rate and the increase in the incidence of symptomatic cholelithiasis in children. Please comment on this in the introduction. [Gallbladder Disease in Children: A 20-year Single-center Experience. Indian Pediatr. 2019;56(5):384-386.].

3. Please indicate whether the Institutional Review Board has approved the study. Please also provide the approval number and date of approval.

4. The primary and secondary outcomes of the study should be clearly stated in the methodology.

5. Please state the exact inclusion and exclusion criteria of the study in the methodology.

6. As this is a retrospective study, I wonder why patients after 2020 were not included in this study. The inclusion of patients from later years would significantly improve the sample size and the quality of the study. In addition, the exact study period (including dates and months) should be specified.

7. All abbreviations used in a table should be explained in a table legend (e.g. LOS).

8. The presentation of age groups is quite unusual. I would suggest replacing ''30≤Age≤49'' with ''30-49''. The same applies to all age groups in the text and in the tables.

9. The discussion is largely a repetition of results and needs to be significantly improved. The discussion lacks comparison with similar studies from the literature. Do not discuss your results piecemeal. Focus on the results of the main objectives of the study. Write in four consecutive paragraphs (without headings): (i) Summary (no data) of the results of this study; (ii) logical and coherent comparison with existing literature focusing on the main objective(s); (iii) limitations of the study; and (iv) implications for practice/policy/research with a concluding statement.

10. The limitations of the study were not even mentioned.

Comments on the Quality of English Language

The text would benefit from professional editing.

Reviewer 2 Report

Comments and Suggestions for Authors

Scala and Improta conducted a single-center study to analyze the effectiveness of the Lean Six Sigma (LSS) approach in reducing postoperative hospital stay in patients undergoing laparoscopic cholecystectomy. The authors included a group of 478 patients. It was shown that the use of LSS resulted in a significant reduction of postoperative hospitalization time in selected groups of patients. However, the article contains a number of significant errors that need to be corrected before it can be reconsidered for acceptance.

Abstract:

1) The authors did not determine the population size. Please correct.

2) The abbreviation DMAIC is unabbreviated. Please expand the abbreviation.

Introduction:

1) The introduction is too extensive and contains a number of unnecessary sentences. In my opinion, there is no need to explain what a gallbladder is or how gallstones develop. The introduction should introduce the reader to the topic covered in the study. In this case, one should only focus on explaining the cholecystectomy procedure, its advantages over the conventional (open) method, complications as factors influencing the extension of the length of stay and the importance of tools aimed at improving the quality of care while minimizing costs.

2) Lack of a clearly defined purpose of the study, which should always be included at the end of the introduction (usually in the form of the last sentence of the introduction, starting with the statement: the purpose of this study was... or this study was aimed at...)

Material and Methods:

1) Was the study retrospective or prospective?

2) In the methods section, the authors mention 254 patients in the pre-LSS group and 226 after LSS, while in the results they mention 254 and 224 patients, respectively. Please explain the discrepancies.

Results:

1) Why was the hospitalization time so long in the authors' center? In my experience, the absolute majority of patients can be discharged on the first or second postoperative day. What was the reason for such a long hospitalization time, both before and after the implementation of LSS?

2) I would suggest including a table summarizing both study groups in the results. Statistical comparison of variables between groups would also be appropriate.

Discussion:

1) The discussion is a repetition of the results. Moreover, there is only one reference to literature in the discussion, which is unacceptable. The discussion does not bring anything new and does not explain how the results obtained by the authors compare to the results of similar studies in the world literature. The discussion requires a complete transformation and expansion to include at least a few references.

Conclusions:

1) The conclusions are a repetition of the results. Conclusions should be drawn on the basis of the obtained research results, but cannot be a repetition of them.

Ethical issues:

1) The authors mention that IRB approval was not required. I disagree with this statement. Currently, for both prospective and retrospective studies based on living organisms or their tissues, consent or at least an opinion of the appropriate ethics committee should be obtained.

Round 2

Reviewer 1 Report

Comments and Suggestions for Authors

The authors responded appropriately to my comments and significantly improved the manuscript. I have no further comments or requests. The manuscript is acceptable for publication in present form.

Comments on the Quality of English Language

--

Reviewer 2 Report

Comments and Suggestions for Authors

The authors responded to all comments and suggestions. I have no further comments.